# New HSV-1 Anti-Viral 1′-Homocarbocyclic Nucleoside Analogs with an Optically Active Substituted Bicyclo[2.2.1]Heptane Fragment as a Glycoside Moiety

**DOI:** 10.3390/molecules24132446

**Published:** 2019-07-03

**Authors:** Constantin I. Tănase, Constantin Drăghici, Anamaria Hanganu, Lucia Pintilie, Maria Maganu, Alexandrina Volobueva, Ekaterina Sinegubova, Vladimir V. Zarubaev, Johan Neyts, Dirk Jochmans, Alexander V. Slita

**Affiliations:** 1National Institute for Chemical-Pharmaceutical Research and Development, Department of Bioactive Substances and Pharmaceutical Technologies, 112 Vitan Av., 031299 Bucharest-3, Romania; 2Organic Chemistry Center “C.D.Nenitescu”, Spectroscopy Laboratory, 202 B Splaiul Independentei, 060023 Bucharest, Romania; 3Department of Virology, Pasteur Institute of Epidemiology and Microbiology, 197101 St. Petersburg, Russia; 4KU Leuven Department of Micobiology, Immunology and Transplantation, Rega Institute, Laboratory of Virology and Chemotherapy, Herestraat 49, BE-3000 Leuven, Belgium

**Keywords:** 1′-homocarbonucleosides, bicyclo[2.2.1]heptane nucleosides, 6-chloropurine, 6-substituted adenine nucleosides, antiviral activity, herpesviruses, molecular docking

## Abstract

New 1′-homocarbanucleoside analogs with an optically active substituted bicyclo[2.2.1]heptane skeleton as sugar moiety were synthesized. The pyrimidine analogs with uracil, 5-fluorouracil, thymine and cytosine and key intermediate with 6-chloropurine (**5**) as nucleobases were synthesized by a selective Mitsunobu reaction on the primary hydroxymethyl group in the presence of 5-endo-hydroxyl group. Adenine and 6-substituted adenine homonucleosides were obtained by the substitution of the 6-chlorine atom of the key intermediate **5** with ammonia and selected amines, and 6-methoxy- and 6-ethoxy substituted purine homonucleosides by reaction with the corresponding alkoxides. No derivatives appeared active against entero, yellow fever, chikungunya, and adeno type 1viruses. Two compounds (**6j** and **6d**) had lower IC_50_ (15 ± 2 and 21 ± 4 µM) and compound **6f** had an identical value of IC_50_ (28 ± 4 µM) to that of acyclovir, suggesting that the bicyclo[2.2.1]heptane skeleton could be further studied to find a candidate for sugar moiety of the nucleosides.

## 1. Introduction

Nucleosides are a chemical class of active substances used as efficient antitumor [1,2,3] or antiviral drugs [4,5,6,7]. The resistance which appears after prolonged use and their toxicity are two principal secondary effects that request the development of newer, more safe candidates. An impressive number of modifications of the sugar moiety or replacements with carbocyclic fragment, replacement of the base or both have been done and some of the corresponding molecules had antiviral or anticancer activity and became useful drugs [2,3,6,7].

Another direction pursued in search for new active nucleosides was to introduce a methylene group between the heterocyclic base and the sugar moiety. The new compounds are, in fact, acyclic nucleosides, but in the literature, their most common name is “1′-homonucleosides”. The insertion of the methylene group alters the chemical and biological characteristics of the molecules, as follows: (1) The anomeric carbon atom C1′ in a tetrahydrofuran moiety has no more reactivity and the compounds are resistant to the enzyme hydrolysis of C1′-CH_2_Base bond, (2) the methylene group allows free rotation between the base and sugar moiety, increasing the flexibility of the molecule in the enzyme pocket(s), (3) electronic and steric interactions between the base and sugar moiety are decreased, (4) the lipophilicity of the compounds is slightly increased, a factor sometimes important for the transport of the molecule across the cell wall, (5) the separation between HO-C5′ of the (un)modified sugar and the N1 or N9 nitrogen atoms of the heterocyclic bases is slightly increased and many times this is the cause of the biological activity or of the lack of activity of the analogs [8].

In the 1′-homonucleoside, the biological activity mainly decreased compared with that of the parent nucleoside [9,10]. The substitution of oxygen with nitrogen [11] or sulfur [12] did not lead to active compounds. However, some changes in the sugar moiety have led to carbocyclic compounds with potent antiviral or anticancer activity.

The introduction of a *cyclopropyl* fragment led to the 1′-homocarbanucleosides **I** [13] with “extremely potent antiherpetic activity” [14] and **II**, with anti-VZV activity, IC_50_ = 0.027 µg/mL [14] (Figure 1). It is interesting that the introduction of a double bond between the cyclopropyl ring and the methylene group, linked to the N-9 nitrogen, in a *Z*-conformation of the hydroxymethyl and nucleobase, gave potent active compounds **III** and **IV** against HCMV (IC_50_ = 1–2.1 μM and 0.04–2.1 μM) and EpsteinBarr virus (IC_50_ = 0.2 μM and 0.3 μM) (Figure 1) [15,16].

1′-Homocarbonucleosides with a *cyclobutene* ring had no significant activity against HIV-1 and HSV-1 [17]. In the series of 2′- or 3′-hydroxymetyl*cyclopentane*-1′-homocarbanucleosides, a few compounds presented a slight antiviral or anticancer activity [18,19], while 1′,3′-disubstituted cyclopentene analogs [20] or 2′,3′-*cis* diols [21], were found to be inactive. Other analogs synthesized had low or no antiviral activity [22,23,24], with the exception of the adenine analog [24]. However, a few chemical structures were fruitfully used to obtain active 1′-homocarbanucleosides (Figure 2), like for example:-**V**, with a 2,2,3-trimethylcyclopentanol, active against HIV-1 and HIV-2 at an EC_50_ = 4–14 µg/mL [25,26]. Some of these compounds exhibited considerable cytostatic activity, on Molt4/C8 (IC_50_ = 13.2 and 3.8 µg/mL with nucleobase 6-*Cl*-2-amino-purine and 2,6 diamino-purine) and L1210/0 (IC_50_ = 16.2 and 13.9 µg/mL with nucleobase 6-*Cl*-2-amino-purine and 2-chloro-purine) cancer lines [25,26]. It is interesting that compound **VI** with the opposite conformation of the cyclopentane ring is inactive (Figure 2) [27].-**VII**, with a 3-hydroxymethyl-indane scaffold, most active being on L1210/0, Molt4/C8 (IC_50_ = 1.4–5.7 µg/mL) and CEM (IC_50_ = 0.63–3.3 µg/mL) cancer cell lines [28,29].-**VIII** and **IX**, with a cyclopenta[*c*]pyrazole moiety [30] or cyclopenta[*d*]pyrazole moiety and hydroxymethyl protected as TBDMS [31,32]. Compounds **VIII**, with adenine or 6*-Cl*-2-aminopurine as the base, are more active against VZV/TK^-^ strain (EC_50_ = 1.5 and 2.1 µmol) than acyclovir, used as reference (EC_50_ = 27 µmol) [30]. The same compounds are as active as ganciclovir (EC_50_ = 0.25 and 0.40 µmol for both compounds) against cytomegalovirus AD169 and DAVIS 07/1 strains (EC_50_ = 0.50 and 0.50 µmol for adenine analog and EC_50_ = 0.44 and 0.39 µmol for 6-*Cl*-2-amino-purine analog). Compounds **IX**, with 6-*Cl*-Pu, 6-*Cl*-2-amino-Pu, Uracil, and with the opposite position of N-Me, present significant anticancer activity on L1210/0, Molt4/C8 and CEM cell lines (IC_50_ = 3.2–11 µg/mL) [8,31,32].

In our previous papers [33,34,35,36,37,38] we used an optically active bicyclo[2.2.1]heptane scaffold to obtain a library of L-type carbocyclic nucleosides **X** and tested them for antiviral and anticancer activity (Figure 3). Compounds **Xa** and **Xb** were “the most prospective for their antiviral activity against influenza virus due to their low toxicity and high activity” [35]. Compound **Xc** was very active against coxsackievirus B4, with EC_50_ = 0.6 µg/mL and selective index of 141, compound **Xb** being also promising. The compounds had low anticancer activity.

The promising results with the specified substituted constrained bicyclo[2.2.1]heptane scaffold, gave us the idea to use it for obtaining 1′-homocarbanucleosides, linking the nucleobase to the hydroxymethyl group and keeping the 5-*endo*-OH group free, in search to obtain potentially antiviral compounds. The works in this direction are presented below.

## 2. Results and Discussion

### 2.1. Molecular Design

The docking studies were performed using CLC Drug Discovery Workbench Software (professional software). The score and hydrogen bonds formed with the amino acids from the group interaction atoms were used to predict the binding modes, the binding affinities and the orientation of the docked ligands in the active site of the protein receptor. The protein-ligand complex was realized based on the X-ray structure of herpes simplex type-1 thymidine kinase (TK) in complex with acyclovir (AC2), who was downloaded from the Protein Data Bank (PDB ID: 2KI5, for a 1.9Å resolution) [39]. The TK enzyme was studied for binding capabilities in the development of new, more effective HSV-1 inhibitors in many papers (for example [40]). The binding site and binding pockets, used to guide the molecular docking of the ligands, were established, and the search was carried out inside the binding site volume (the green sphere, Appendix A).

Firstly, the binding site and docking pose of the co-crystallized AC2 interacting with amino acids residues are shown in Appendix A. Then hydrogen bonds between amino acids residues of thymidine kinase and co-crystallized AC2 and docking validation (for the co-crystallized) were done (see Appendix A). The new 1′-homocarbacyclonucleosides were finally docked, and the results of the calculated properties are presented in Table 1 (flexible bonds, Lipinski violations, the number of hydrogen bond donors, the number of hydrogen bond acceptors and log P). These parameters can predict if a molecule possesses properties that might turn it into an active drug, according to the Lipinski’s rule of five. The number of violations of the Lipinski rules allows to evaluate drug-likeness for a molecule. According to the data presented in Table 1, all of the compounds comply with the Lipinski rules (Lipinski violation is 0) [41]. In Table 1, the results of the docking study (docking score, RMSD < 2Å) are also presented.

The docking score (PLANT_PLP_ score) is a function described in Korb et al. [42]. For a strong binding, the score has a negative value, for weak or non-existing binding the score has a less negative or even positive value.

Additionally, group interaction, hydrogen bonds of ligands with amino acid residues were determined and hydrogen bond length was calculated. These are presented in the Appendix A.

The data presented in the Table 1 and Appendix A and Figure 4 show that compounds **4a**–**4d**, **5**, **6a**–**6b**, **6e**, **6i**–**6k**, **7a**–**7b** have a docking score greater than that of acyclovir (−49.29, RMSD 0.71).

Compounds **6i** and **6k** presented the best scores, −70.21, respectively −70.07. The docking pose of the two ligands interacting with the amino acids residues is presented in Appendix A. In fact, all new ligands (1′-homocarbanucleosides) were found to have the same orientation as AC2 (Appendix A), as it can be observed in Appendix A for linking to a group of interaction with the amino acids.

The orientation and the docking pose of AC2 with the pyrimidine ligands **4a**–**4d**, **6a**–**6k** N^6^-substituted adenine ligands and N^6^-alkoxy-purine **7a**–**7b** ligands are presented in Appendix A, and show a good overlay.

The docking score was correlated with the experimental HSV-1 antiviral results (IC_50_), and the best matching is for compound **6j** (score −62.08, RMSD 1.57, IC_50_ = 15 ± 2 µM) (Figure 5). The second compound (**6d**) with lower IC_50_ (21 ± 4 µM) than acyclovir (28 ± 4 µM) had a little lower value for the docking score (−39.94, RMSD 0.13) than that for acyclovir (−49.29, RMSD 0.71).

The compounds with the best scores: **6k** and **6i** (−70.07, RMSD 0.09, respectively, −70.21, RMSD 0.07), presented greater values for IC_50_ (48 ± 6 and 155 ± 13 µM) which are not in agreement with the experimental anti-HSV-1 activity. Therefore, the molecular design should be taken with caution, because the results are prediction values, not experimental ones.

### 2.2. Chemistry

The synthesis of these new 1′homocarbanucleosides started from the diol **2**, obtained as a major isomer by sodium borohydride reduction of the optically active keto-alcohol intermediate **1** by our previous procedure [36]. The bulk of 5-endo-OH isomer **2** was obtained in pure form by crystallization, in yield greater than 77%. By low pressure chromatography (LPC) purification of the mother liquors, the total yield was increased to 91% (Scheme 1).

We chose to use a Mitsunobu reaction to alkylate the pyrimidine bases with the primary hydroxyl group of the diol **2**, as the most direct chemical option to obtain the 1′-homocarba-pyrimidine nucleosides **4** (Scheme 1). For this goal, we relied on the fact that in the Mitsunobu reaction, the primary hydroxyl group has greater reactivity than the secondary hydroxyl group, linked endo to C_5_ carbon atom. Certainly, the protection of the secondary hydroxyl group would lead to greater yields of the final products **4**, but this would make the sequence of reactions longer, and at that time the yield was not considered so important. Compounds **4a**, **4b,** and **4c** were obtained in 20.3, 34.5 and respectively 36.6% yields. Compound **4d** was obtained in 23.5% yield by the Mitsunobu reaction of diol **2** with N^4^-Cytosine benzoate, followed by the deprotection of the benzoate group on crude alkylation reaction product by transesterification (MeONa/MeOH). The isolation of the pure compound **4d** was realized by LPC. The reduced yield for the pyrimidine analogs is attributed to the formation of O^2^,O^4^- and N^1^,O^4^-bis-akylated secondary compounds, as we observed previously [43] for the alkylation of the diol **2** protected at the primary hydroxyl as benzoate. In this case, alkylated compounds to both hydroxyls of the diol **2** could also be formed. All secondary compounds had higher mobility on TLC, in the domain of triphenylphosphine oxide as R_f_, and were not isolated pure for characterization.

The second part of the paper is focused on obtaining 1′-homocarbanucleoside analogs with a purine base. In this case, we also chose the most direct synthesis, which implies two steps. In the first step, we obtained the key optically active intermediate **5** in 67.6% yield by a slight modification of the Mitsunobu reaction of 6-chloropurine [44] with our diol **2** (Scheme 2). In the second step, the 6-chlorine atom of the key intermediate **5** was substituted with ammonia to obtain the adenine analog **6a**, and with selected amines, used previously in our papers, where 6-chloropurine was linked *exo* to the C_5_ carbon atom [33,34], as in **X** (Figure 3, R = Cl), to 1′-homocarba-N^6^-substituted adenine nucleosides **6a**–**6k**, in good yields (from 77.9% for **6k** to 95.1% for **6d**). Only the compound **6g** was obtained in a moderate yield of 49.2%. The chlorine atom was also substituted by a methoxy (**7a**) or ethoxy group (**7b**), by the reaction of the key intermediate **5** with sodium methoxide in methanol or sodium ethoxide in ethanol in good yield (70.9%, respectively 83.6%).

The pure compounds were fully characterized and have been screened for antiviral activity.

### 2.3. ^1^H, ^13^C-NMR, MS, Elemental Analysis and Optical Rotation

All new compounds were purified by pressure chromatography and analyzed by optical rotation, IR, ^1^H-, ^13^C-NMR, and 2D-NMR spectra, presented at the experimental part. The analytical data were in full agreement with the proposed structures. In ^1^H-NMR spectra, the assignment was performed on the basis of chemical shifts, signal intensity and multiplicity of H-H coupling constants. ^13^C-NMR and complementary 2D-NMR and decoupling spectra gave the correct signal for each proton and carbon atom in the molecules. The ^1^H-, ^13^C-, and 2D-NMR spectra of the key intermediate **5** and of the new 1′homocarbanucleosides **4**, **6** and **7** are presented in the Appendix A.

### 2.4. Antiviral Activity of the Compounds

The new 1′-homocarbanucleosides analogs were tested on different viruses:1)adeno-, herpes- and influenza viruses, at Pasteur Institute of Epidemiology and Microbiology, Department of Virology, St. Petersburg, Russia, and2)enteroviruses type: enterovirus 71 (EV71), yellow fever and Chikungunya viruses, at Rega Institute, Laboratory of Virology and Chemotherapy, Leuven-Belgium

The results of anti-viral testing of novel 1′-homocarbanucleosides analogs against adeno-, influenza, and herpesvirus type 1are summarized in Table 2.

As can be seen from the data presented, no derivatives appeared active against adenovirus. Only one compound demonstrated moderate inhibiting activity against influenza virus while 5 out of 18 compounds were active against the herpes virus. Importantly, there was no correlation between activity against influenza virus and herpes virus. Indeed, the cytosine compound **4d**, the only derivative active against influenza virus, was ineffective against HSV-1. On another hand, the compounds active against herpes (5-chloro-purine **5**, and 6 alkyl-substituted **6c**, **6d**, **6f**, **6j**) did not show the same activity against influenza virus. These differences may reflect the specificity in their targets. With high probability, the targets for nucleoside analogs should be the enzymes participating in the nucleic acids metabolism, in particular, viral polymerases. Herpes virus and influenza virus have a different organization of their genomes. The former has double-stranded DNA genome while the latter—single strand RNA genome of negative polarity. Due to differences in genomes of influenza virus and herpes virus, one can suggest that different inhibitors, although belonging to one chemical class, are needed to interfere with the activities of two different enzymes.

It is known that the best-known anti-herpes compound acyclovir is phosphorylated in infected cells by viral thymidine kinase following by incorporation of its triphosphate form into the DNA resulting in chain termination and death of infected virus cells [45]. Probably, the compounds we studied have a similar mechanism of activity, and for this reason, are ineffective against influenza virus who does not encode its own kinase. Further experiments are needed for deciphering the mechanism of activity and identifying the target(s) of lead compounds, including but not limited to time-of-addition experiments, virus yield reduction assay as well as tests for specific viral enzymes and selection and analysis of resistant mutants.

By analyzing the structures of active compounds, one can infer that even minor differences in the substituents **R** result in a dramatic change in the anti-viral properties. Indeed, the similar compounds **6f** and **6g** differing in one nitrogen bridge atom demonstrate almost a 30-fold difference in the virus-inhibiting activity against HSV-1. The introduction of an amino group between 4-methylpyperazine and N^6^ nitrogen atom resulted in the complete loss of virus inhibition in **6g** compared to **6f**. Change of the hexane group to unsaturated phenyl group (**6d** to **6k**) resulted in the decrease of virus inhibition (IC_50_ from 21 to 48 µM), while the addition of the metoxy group (**6j**) restored the activity (IC_50_ 15). Even the change of cyclohexyl for cyclo-pentyl group (**6d** to **6c**) decreased the virus inhibition (IC_50_ from 21 to 47 µM). Chorine- and amino- substituted derivatives **5** and **6a** did not differ significantly in their properties. Taking into account only the IC_50_ of the new 1′-homocarbacyclonucleosides in comparison with that of acyclovir (28 ± 4 µM), compounds **6j** and **6d** have a lower IC_50_ (15 ± 2 and 21 ± 4 µM), compound **6f**—an identic value (28 ± 4 µM) and another three compounds had the values of IC_50_ about twice higher. Five of them have also a SI greater than 10 (See Table 3). Importantly, compounds **6f**, **6d**, and **6j** showed values of IC_50_ against herpes virus similar to that of acyclovir, but their cytotoxicity was higher. This suggests that further optimization of the structure directing to decrease the toxicity could provide good pharmacologic characteristics of the resulting compounds.

No compounds were active against yellow fever and Chikungunya viruses. Compounds **6c**, **6d**, **6j**, and **6k** presented low activity against EV 71 virus (IC_50_ 20–80 µM) (Table 3).

Taken together, our findings suggest that the bicyclo[2.2.1]heptane skeleton could be a candidate for sugar moiety of the nucleosides with potential virus-inhibiting properties.

## 3. Materials and Methods 

### 3.1. Experimental-Chemistry 

Melting points (uncorrected) were determined in open capillary on an OptiMelt melting point apparatus (MPA 100, Stanford Research System, Inc., Sunnyvale, CA, USA). The progress of the reaction was monitored by TLC on silica gel 60 or 60F_254_ plates (Merck, Darmstadt, Germany) eluted with the solvent systems: I dichloromethane-methanol, 9:1, II dichloromethane-methanol, 95:5, III dichloromethane-methanol, 4:1. Spots developed in UV and with 15% H_2_SO_4_ in MeOH (heating at 110 °C, 10 min). IR spectra were recorded on FT-IR-100 Perkin Elmer spectrometer (Perkin Elmer, Shelton, CT, USA), in solid phase by ATR and frequencies were expressed in cm^−1^, with the following abbreviations: w = weak, m = medium, s = strong, v = very, br = broad. ^1^H-NMR and ^13^C-NMR spectra are recorded on Bruker Fourier 300 MHz spectrometer (300 MHz for ^1^H and 75 MHz for ^13^C, Karlsruhe, Germany), or Bruker Avance III 500 MHz spectrometer (500 MHz for ^1^H and 125 MHz for ^13^C), spectrometer chemical shifts are given in ppm relative to TMS as internal standard. Complementary spectra: 2D-NMR and decoupling were done for the correct assignment of NMR signals. The numbering of the atoms in the compounds is presented in schemes. Diol **2** was obtained by sodium borohydride reduction of the keto group of compound **1,** as previously presented [36], having mp 116–117.2 °C, [α]_D_ = 11.3° (1% MeOH).

#### 3.1.1. Synthesis of 1-(((1*S*,2*S*,4*S*,7*R*)-2-Chloro-5-hydroxybicyclo[2.2.1]heptan-7-yl)methyl)-5-fluoropyrimidine-2,4(1*H*,3*H*)-dione, **4a**

Triphenylphosphine (Ph_3_P) (5.246 g, 20 mmol) and 5-fluorouracil (2.62 g, 20 mmol) were suspended in 120 mL anh tetrahydrofuran and stirred at rt for 20 min under anh. argon atmosphere, then cooled to 0 °C with an ice-water bath. DIAD (4.16 mL, 20 mmol) was added dropwise. After one hour, a solution of diol **2** (1.766 g, 10 mmol) in tetrahydrofuran (80 mL) was added during 50 min and stirred overnight monitoring the end of the reaction by TLC (I, R_f **2**_ = 0.43, R_f_ = 0.55). A secondary compound with R_f_ = 0.63 was also formed in the reaction and was not isolated pure. Solvent was distilled under reduced pressure, the residue was purified by low pressure chromatography (LPC) (solvent system, I), resulting a pure fraction of 587 mg (20.3%) **4a**, mp 212.3–213.7 °C (EtOH), [α]_D_ = 39.4° (1% EtOH), IR: 3377s, 3170w, 3045w, 2962w, 2878w, 2819w, 1762w, 1703s, 1663vs, 1475w, 1434w, 1375m, 1339m, 1237s, 1113m, 1077m, 1006m, 900w, 817m, ^1^H-NMR (DMSO-*d*_6_, δ ppm, *J* Hz): 11.77 (d, 1H, NH, 4.8), 8.10 (d, 1H, H-6, *J_H-F_* = 2.8 Hz), 4.73 (brs, 1H, OH) deuterable, 4.21 (dd, 1H, H-8, 11.0, 14.0), 4.08 (brdd+TFA, 1H, H-2, 3.0, 8.1), 3.92 (brd+TFA, 1H, H-5, 9.6), 3.62 (dd, 1H, H-8, 3.6, 14.0), 2.66 (dd, 1H, H-3, 8.0, 13.7), 2.26 (d, 1H, H-1, 4.5), 2.13 (s, 1H, H-4), 2.11 (brd, 1H, H-3, 13.4), 2.01–1.94 (m, 2H, H-6, H-7), 0.79 (dd+TFA, 1H, H-6, 2.2, 13.4), ^13^C-NMR (DMSO-*d*_6_, δ ppm): 157.48 (d, C-4′, *J* = 24.8 Hz), 149.72 (C-2′), 139.67 (d, C-5, *J* = 227.3 Hz), 130.13 (d, C-6, *J* = 33 Hz), 68.11 (C-5), 61.48 (C-2), 48.76 (C-1) from COSY and HMBC, 47.99 (C-7), 44.37 (C-8), 45.01 (C-4), 38.78 (C-6), 32.14 (C-3).

#### 3.1.2. Synthesis of 1-(((1*S*,2*S*,4*S*,5*R*,7*R*)-2-Chloro-5-hydroxybicyclo[2.2.1]heptan-7-yl)methyl)pyrimidine-2,4(1*H*,3*H*)-dione, **4b**

Starting from 10 mmol diol **2** and uracil (2.24 g, 20 mmol) as in the example 3.1, 935 mg (34.5%) **4b**, were obtained as a foam, mp 202.2–204.5 °C (EtOH), [α]_D_ = 35.0° (1% EtOH), **4b**, IR: 3353m, 3194w, 3103w, 2968w, 2881w, 1700s, 1657vs, 1461m, 1337m, 1241m, 1073m, 1007w, 814m, 612m, ^1^H-NMR (DMSO-*d*_6_, δ ppm, *J* Hz): 11.25 (1H, NH), 7.68 (d, 1H, H-6′, 7.8), 5.55 (d, 1H, H-5′, 7.8), 4.73 (d, 1H, OH, 3.7), 4.20 (dd, 1H, H-8, 10.4, 13.8), 4.09 (brd, 1H, H-2, 7.9), 3.92 (brd, 1H, H-5, 5.8), 3.70 (dd, 1H, H-8, 4.2, 13.8), 2.70 (dd, 1H, H-3, 7.9, 14.0), 2.26 (d, 1H, H-1, 4.5), 2.11 (s, 1H, H-4), 2.09 (brd, 1H, H-3, 14.0), 2.02–1.93 (m, 2H, H-6, H-7), 0.79 (dd, 1H, H-6, 2.3, 13.3), ^13^C-NMR (DMSO-*d*_6_, δ ppm): 163.75 (C-4′), 151.06 (C-2′), 145.70 (C-6′), 100.96 (C-5′), 68.06 (C-5), 61.44 (C-2), 48.59 (C-1), 48.14 (C-7), 46.09 (C-8), 45.05 (C-4), 38.82 (C-6), 32.18 (C-3).

#### 3.1.3. Synthesis of 1-(((1*S*,2*S*,4*S*,7*R*)-2-Chloro-5-hydroxybicyclo[2.2.1]heptan-7-yl)methyl)-5-methylpyrimidine-2,4(1*H*,3*H*)-dione, **4c**

Starting from 10 mmol diol **2** and tymine (2.522 g, 20 mmol) as in the example 3.1, 1.043 g (36.6%) **4c**, were obtained as a foam, R_f_ = 0.50 (I). A sample was crystallized from EtOH, mp 211.3–212.0 °C (dec.), [α]_D_ = 43.2 ° (1% EtOH), IR: 3390s, 3188m, 3064m, 2891m, 2815w, 1679vs, 1660m, 1335m, 1249w, 1125w, 1004m, 833m, ^1^H-NMR (DMSO-*d*_6_, δ ppm, *J* Hz): 11.23 (1H, NH), 7.56 (s, 1H, H-6′), 4.72 (d, 1H, OH, 3.4), 4.20 (dd, 1H, H-8, 10.8, 13.9), 4.09 (brd, 1H, H-2, 4.5, 8.0), 3.91 (brd, 1H, H-5, 6.1), 3.64 (dd, 1H, H-8, 3.3, 13.9), 2.67 (dd, 1H, H-3, 8.0, 14.7), 2.26 (d, 1H, H-1, 4.5), 2.12–2.10 (m, 2H, H-4, H-3), 2.02–1.92 (m, 2H, H-6, H-7), 1.75 (s, 3H, CH_3_), 0.79 (dd, 1H, H-6, 2.2, 13.3), ^13^C-NMR (DMSO-*d*_6_, δ ppm): 164.33 (C-4′), 151.00 (C-2′), 141.46 (C-6′), 108.60 (C-5), 68.10 (C-5), 61.44 (C-2), 48.67 (C-1), 48.21 (C-7), 45.84 (C-8), 45.03 (C-4), 38.81 (C-6), 32.20 (C-3), 11.98 (CH_3_).

#### 3.1.4. Synthesis of 4-Amino-1-(((1*S*,2*S*,4*S*,7*R*)-2-chloro-5-hydroxybicyclo[2.2.1]heptan-7-yl)methyl)pyrimidin-2(1*H*)-one, **4d**

Starting from 5 mmol diol **2**, with Ph_3_P (10 mmol, 2.623 g), N^4^-Cytosine benzoate (10 mmol, 2.152 g), DIAD (10 mmol, 2.1 mL) in THF (90 mL), as in the example 3.1, followed by the cleavage of the benzoate group on the crude product in 1 M MeONa (100 mL) and methanol (100 mL) overnight and LPC (I), 317 mg (23.5%) **4d** were obtained as a foam, R_f **4d**_ = 0.30, [α]_D_ = 33.2° (1% EtOH), IR: 3532m, 3335s, 3209m, 3100m, 2959m, 2874w, 1629vs, 1605vs, 1523m, 1480vs, 1436s, 1381s, 1270s, 1196m, 1074m, 994m, 784m, ^1^H-NMR (DMSO-*d*_6_, δ ppm, *J* Hz): 7.61 (d, 1H, H-6′, 7.1), 7.01 (s, 2H, NH), 5.64 (d, 1H, H-5′, 7.1), 4.72 (d, 1H, OH, 3.2), 4.15 (dd, 1H, H-8, 10.0, 13.5), 4.09 (brdd, 1H, H-2, 3.1, 8.0), 3.88 (brd, 1H, H-5, 6.5), 3.69 (dd, 1H, H-8, 4.2, 13.5), 2.66 (dd, 1H, H-3, 8.0, 13.5), 2.23 (d, 1H, H-1, 4.7), 2.11 (brd, 1H, H-3, 13.5), 2.09 (s, 1H, H-4), 2.00–1.93 (m, 2H, H-6, H-7), 0.78 (dd, 1H, H-6, 1.9, 13.1), ^13^C-NMR (DMSO-*d*_6_, δ ppm): 165.89 (C-4′), 155.85 (C-2′), 146.04 (C-6′), 93.22 (C-5′), 68.13 (C-5), 61.54 (C-2), 48.49, 48.38 (C-1, C-7), 47.18 (C-8), 45.00 (C-4), 38.82 (C-6) in DMSO, see HETCOR, 32.30 (C-3).

#### 3.1.5. Synthesis of (1*S*,4*S*,5*S*,7*R*)-5-Chloro-7-((6-chloro-9*H*-purin-9-yl)methyl)bicyclo[2.2.1]heptan-2-ol, **5**

Compound **5** was synthesized by the slightly modified procedure used in example 3.1.1: Diol **2** (3.53 g, 20 mmol) was dissolved in tetrahydrofuran (160 mL), and the solution was added during one hour to the complex formed by the addition of DIAD (8.3 mL, 40 mmol) to a solution of 6-chloropurine (10.49 g, 40 mmol) and Ph_3_P (6.18 g, 40 mmol) in tetrahydrofuran (280 mL) at −6 °C over 40 min, followed by a one-hour waiting time for the complete formation of the betaine complex. The reaction mixture was stirred over weekend, monitoring the end of the reaction by TLC (I, R_f **6-ClPu**_ = 0.39, R_f **2**_ = 0.43, R_f **5**_ = 0.55), the solvent was removed under reduced pressure and the residue was purified by LPC (solvent system I). The compound was crystallized from CH_2_Cl_2_-hexane, resulting 4.24 g (67.7%) of crystallized **5**, mp 198.0–199.0 °C (>200.5 °C dec.), [α]_D_ = 7.35° (1% EtOH), IR: 3394m, 3312m, 3076w, 2956w, 2899w, 1596s, 1563s, 1444m, 1402m, 1335vs, 1262m, 1212m, 1168m, 1079m, 1045m, 935s, 640m, ^1^H-NMR (DMSO-*d*_6_, δ ppm, *J* Hz): 8.79 (s, 1H, H-8′), 8.77 (s, 1H, H-2′), 4.74 (d, 1H, OH, 3.1), 4.73 (dd, 1H, H-8, 11.0, 14.2), 4.49 (dd, 1H, H-8, 5.0, 14.2), 4.16 (brdd, 1H, H-2, 3.1, 7.6), 3.86 (m, 1H, H-5), 2.75 (dd, 1H, H-3, 7.6, 14.2), 2.36 (d, 1H, H-1, 4.5), 2.30–2.22 (m, 2H, H-3, H-7), 2.05–1.94 (m, 2H, H-6, H-4), 0.82 (dd, 1H, H-6, 2.0, 13.3), ^13^C-NMR (DMSO-*d*_6_, δ ppm): 151.98 (C-6′), 151.52 (C-2′), 148.99 (C-4′), 147.44 (C-8′), 130.73 (C-5′), 67.89 (C-5), 61.25 (C-2), 48.48 (2C, C-1, C-7), 45.32 (C-8), 42.64 (C-4), 38.84 (C-6), 32.16 (C-3).

##### General Procedure for Synthesis of Carbocyclic 1′-Homonucleosides **6a**–**7k**

The key intermediate **5** (0.4 or 0.8 mmol) was stirred with the specified amount of amine, without or in the presence of ethanol (3.5 or 7 mL) as solvent, without or in the presence of triethylamine, for the time frame mentioned in each example. The solvent (ethanol) and the volatile amines were removed under reduced pressure, the residue was taken in CH_2_Cl_2_, washed with water, organic phase dried (Na_2_SO_4_) and concentrated to dryness (the aqueous phases were extracted with CH_2_Cl_2_, and unified with the crude product). The crude product was purified by LPC and the pure compound was obtained as foam or waxy. Some of the compounds were crystallized in the solvent(s) specified in each case.

#### 3.1.6. Synthesis of (1*S*,4*S*,5*S*,7*R*)-7-((6-Amino-9*H*-purin-9-yl)methyl)-5-chlorobicyclo[2.2.1]heptan-2-ol, **6a**

Compound **5** (0.8 mmol), 25% NH_4_OH (40 mL), methanol (20 mL) were heated at 80–90 °C in a pressure vessel for 24 h; TLC (I, R_f_ = 0.29), 197 mg (83.8%) of pure **6a** were obtained, mp 223.0–223.5 °C (EtOH), [α]_D_ = 1.3° (0.5% EtOH), IR: 3307s, 3163s, 2991m, 2961w, 2916w, 2880w, 1730w, 1651vs, 1583vs, 1514m, 1452w, 1328m, 1302m, 1241m, 1200w, 1172w, 1076m, 1002m, 894m, 647m, ^1^H-NMR (DMSO-*d*_6_, δ ppm, *J* Hz): 8.19 (s, 1H, H-8′), 8.14 (s, 1H, H-2′), 7.22 (s, 2H, NH_2_), 4.72 (d, 1H, OH, 3.5), 4.59 (dd, 1H, H-8, 10.1, 14.1), 4.31 (dd, 1H, H-8, 5.1, 14.1), 4.15 (dd, 1H, H-2, 3.6, 7.8), 3.87 (m, 1H, H-5), 2.73 (dd, 1H, H-3, 8.1, 14.1), 2.33 (d, 1H, H-1, 4.5), 2.26–2.18 (m, 2H, H-3, H-7), 2.01 (m, 1H, H-4), 1.98 (ddd, 1H, H-6, 5.0, 9.9, 13.5), 0.81 (dd, 1H, H-6, 2.2, 13.5), ^13^C-NMR (DMSO-*d*_6_, δ ppm): 156.42 (C-6′), 152.92 (C-**2**′), 149.96 (C-4′), 141.17 (C-**8**′), 119.03 (C-1′), 68.40 (C-5), 61.76 (C-2), 49.38 (C-7), 48.80 C-1), 45.74 (C-4), 42.14 (C-8), 39.17 in DMSO (C-6), 32.18 (C-3).

#### 3.1.7. Synthesis of (1*S*,4*S*,5*S*,7*R*)-5-Chloro-7-((6-(cyclopropylamino)-9*H*-purin-9-yl)methyl)bicyclo[2.2.1]heptan-2-ol, **6b**

Compound **5** (0.8 mmol), 2 mL cyclopropylamine, 72 h, rt, R_f_ = 0.39). LPC (II) gave 252 mg (94.4%) of pure **6b**, as a foam, [α]_D_ = −15.0° (1% EtOH), IR: 3256brm, 3089w, 3035m, 2962s, 1613vs, 1476m, 1450m, 1348m, 1295m, 1260s, 1082s, 1009vs, 793s, ^1^H-NMR (DMSO-*d*_6_, δ ppm, *J* Hz): 8.25 (s, 1H, H-2′), 8.19 (s, 1H, H-8′), 7.88 (brs, NH), 4.72 (d, 1H, OH, 3.5), 4.60 (dd, 1H, H-8, 10.1, 14.0), 4.32 (dd, 1H, H-8, 5.1, 14.0), 4.14 (brdd, 1H, H-2, 3.3, 8.0), 3.85 (m, 1H, H-5), 3.03 (m, 1H, H-1′′), 2.73 (dd, 1H, H-3, 8.0, 14.1), 2.32 (d, 1H, H-1, 4.4), 2.26–2.20 (m, 2H, H-3, H-7), 2.02–1.93 (m, 2H, H-6, H-4), 0.81 (brd, 1H, H-6, 2.0, 14.0), 0.73–0.60 (m, 4H, CH_2_-Cyclopropyl), ^13^C-NMR (DMSO-*d*_6_, δ ppm): 155.53 (C-6′), 152.39 (C-2′), 148.95 (C-4′), 140.58 (C-8′), 118.95 (C-5′), 67.96 (C-5), 61.33 (C-2), 48.96 (C-7), 48.36 C-1), 45.31 (C-4), 41.70 (C-8), 38.66 in DMSO (C-6), 32.20 (C-3), 23.91 (C-1″), 6.42 (C-2″, C-3″).

#### 3.1.8. Synthesis of (1*S*,4*S*,5*S*,7*R*)-5-Chloro-7-((6-(cyclopentylamino)-9*H*-purin-9-yl)methyl)bicyclo[2.2.1]heptan-2-ol, **6c**

Compound **5** (0.8 mmol), cyclopentylamine (2.2 mL), stirring at rt for 72 h; TLC (I, R_f_ = 0.60). LPC (II) gave 234.5 mg (81.0%) of pure compound **6c** as a foam, [α]_D_ = −18.3° (1% EtOH), IR: 3319brm, 2955m, 2868w, 1613vs, 1534m,1477w, 1333m, 1296m, 1083w, 1002w, 854w, 647w, ^1^H-NMR (CDCl_3_, *δ* ppm, *J* Hz): 8.37 (s, 1H, H-2′), 7.94 (s, 1H, H-8c′), 5.96 (brs, 1H, NH), 4.61 (dd, 1H, H-8, 8.2, 14.3), 4.57 (m, 1H, H-1″) from Hetcor, 4.54 (dd, 1H, H-8, 7.8, 14.3), 4.14–4.11 (m, 2H, H-2, H-5), 2.97 (dd, 1H, H-3, 8.0, 14.4), 2.34 (d, 1H, H-1, 4.5), 2.33–2.28 (m, 2H, H-4, H-7), 2.21–2.02 (m, 4H, H-6, H-4, 2H-2″), 1.80–1.65 (m, 4H, H-3″), 1.56 (m, 2H, H-2″), 0.96 (brd, 1H, H-6, 14.0), ^13^C-NMR (CDCl_3_, *δ* ppm): 155.54 (C-6′), 152.19 (C-2′), 148.74 (C-4′), 139.65 (C-8′), 119.53 (C-5′), 69.24 (C-5), 60.55 (C-2), 52.40 (C-1″), 49.07 (C-7), 48.03 C-1), 45.73 (C-4), 42.59 (C-8), 39.36 (C-6), 33.40 (C-2″), 32.73 (C-3), 23.66 (C-3″).

#### 3.1.9. Synthesis of (1*S*,4*S*,5*S*,7*R*)-5-Chloro-7-((6-(cyclohexylamino)-9*H*-purin-9-yl)methyl)bicyclo[2.2.1]heptan-2-ol, **6d**

Compound **5** (0.8 mmol), cyclohexylamine (4.8 mL), stirring for six days at rt, TLC (I, R_f_ = 0.55). LPC gave 286 mg (95.1%) of pure compound **6d** as a foam, [α]_D_ = 2.9° (1% EtOH), IR: 3320brm, 2926s, 2852m, 1612vs, 1474m, 1446m, 1331m, 1295m, 1253m, 1083m, 1001w, 895w, 646w, ^1^H-NMR (CDCl_3_, *δ* ppm, *J* Hz): 8.35 (s, 1H, H-2′), 7.84 (s, 1H, H-8′), 5.89 (brs, 1H, NH), 4.61 (dd, 1H, H-8, 8.2, 14.4), 4.54 (dd, 1H, H-8, 8.0, 14.4), 4.14–4.11 (m, 2H, H-2, H-5), 2.97 (dd, 1H, H-3, 8.0, 14.5), 2.41–2.28 (m, 3H, H-1, H-4, H-7), 2.17 (brd, 1H, H-3, 14.5), 2.10–2.05 (m, 3H, H-6, 2H-2″), 1.80–1.75 (m, 2H, H-4″), 1.48–1.24 (m, 6H, 2H-2″, 4H-3″), 0.95 (brd, 1H, H-6, 13.2), ^13^C-NMR (CDCl_3_, *δ* ppm): 154.10 (C-**6**′), 153.19 (C-**2**′), 148.83 (C-**4**′), 139.66 (C-**8**′), 119.41 (C-**5**′), 69.17 (C-5), 60.58 (C-2), 50.43 (C-1″), 49.05 (C-7), 48.00 C-1), 45.73 (C-4), 42.57 (C-8), 39.38 (C-6), 33.30 (C-2″), 32.74 (C-3), 26.88 (C-3″), 24.79 (C-4″).

#### 3.1.10. Synthesis of (1*S*,4*S*,5*S*,7*R*)-5-Chloro-7-((6-(((*R*)-1-hydroxy-3-phenylpropan-2-yl)amino)-9*H*-purin-9-yl)methyl)bicyclo[2.2.1]heptan-2-ol, **6e**

Compound **5** (0.4 mmol), phenylalaninol (165 mg, 1 mmol), ethanol (4 mL), Et_3_N (0.3 mL) stirred for 72 h at 60 °C, TLC (I, R_f_ = 0.29). LPC (I) gave 161 mg (94.1%) of pure compound **6e** as a waxy, [α]_D_ = −89.9° (1% EtOH), IR: 3297brs, 2956brm, 1810w, 1695w, 1610vs, 1477m, 1447m, 1330m, 1296m, 1260m, 1079m, 1002m, 896w, 734m, 644w, ^1^H-NMR (CDCl_3_, δ ppm, *J* Hz): 8.32 (s, 1H, H-2′), 7.88 (s, 1H, H-8′), 7.30–7.21 (m, 5H, H-Ar), 6.49 (brs, 1H, NH), 4.60 (dd, 1H, H-8, 7.8, 14.3), 4.57 (m, 1H, H-2″), 4.53 (dd, 1H, H-8, 8.0, 14.3), 4.16–4.10 (m, 2H, H-5, H-2), 3.88 (dd, 1H, H-1″, 2.0, 10.9), 3.74 (dd, 1H, H-1″, 5.4, 10.9), 3.04 (d, 2H, H-3″, 7.1), 2.95 (dd, 1H, H-3, 8.2, 14.7), 2.39 (d, 1H, H-1, 4.3), 2.31–2.27 (m, 2H, H-4, H-7), 2.19 (brdt, 1H, H-3, 14.7), 2.06 (ddd, 1H, H-6, 5.0, 10.3, 13.5), 0.94 (brd, 1H, H-6, 2.5, 13.5), ^1^H-NMR (CDCl_3_ + TFA, δ ppm, *J* Hz): 9.65 (d, 1H, NH, 9.3), 8.68 (s, 1H, H-**2**′), 8.33 (s, 1H, H-**8**′), 7.24–7.16 (m, 5H, H-Ar), 4.85 (dd, 1H, H-8, 7.0, 14.3), 4.78 (dd, 1H, H-8, 8.2, 14.3), 4.61 (m, 1H, H-2″), 4.34 (m, 1H, H-2 or 5), 4.23–4.13 (m, 2H, H-1″), 3.93 (t, 1H, H-5 or 2, 11.4), 3.11 (dd, 1H, H-3″, 5.3, 14.2), 3.02 (dd, 1H, H-3″, 9.0, 14.2), 2.88 (dd, 1H, H-3, 8.0, 15.2), 2.51 (s, 1H, H-1), 2.44 (d, 1H, H-4, 4.2), 2.30 (m, 1H, H-7), 2.26 (brdt, 1H, H-3, 15.2), 2.14 (ddd, 1H, H-6, 4.9, 10.4, 14.4), 1.06 (brd, 1H, H-6, 2.4, 14.4), ^13^C-NMR (CDCl_3_, δ ppm): 154.58 (C-**6**′), 152.72 (C-**2**′), 149.03 (C-**4**′), 140.05 (C-**8**′), 137.89 (Cq-Ar), 129.29 (C-*m* Ar), 128.56 (C-*o* Ar), 126.57 (C-*p* Ar), 119.46 (C-**1**′), 69.35 (C-5), 64.49 (C-1″), 60.49 (C-2), 54.57 (C-2″), 49.03 (C-7), 47.93 C-1), 45.72 (C-4), 42.60 (C-8), 39.44 (C-6), 37.60 (C-3″), 32.74 (C-3).

#### 3.1.11. Synthesis of (1*S*,4*S*,5*S*,7*R*)-5-Chloro-7-((6-(4-methylpiperazin-1-yl)-9*H*-purin-9-yl)methyl)bicyclo[2.2.1]heptan-2-ol, **6f**

Compound **5** (0.8 mmol), *N*-methyl-piperazine (2 mL), ethanol (8 mL), Et_3_N (0.3 mL), stirred for 72 h at rt, TLC (I, R_f_ = 0.35). After work-up, the crude compound was crystallized from CH2Cl2-hexane, and 258.5 mg (74.7%) of pure compound **6f** resulted as needles, mp 172.9–174.0 °C (dec.) [LPC (I) of mother liquors gave another 82 mg of pure **6f** as a foam, total yield: 90.3%], [α]_D_ = 18.4° (1% EtOH), IR: 3317s, 3002m, 2970m, 2933m, 2864m, 2796m, 2471w, 2057w, 1917w, 1657w, 1584vs, 1565vs, 1442s, 1365m, 1294s, 1248s, 1140m, 1078m, 996s, 851w, 646m, ^1^H-NMR (CDCl_3_, δ ppm, *J* Hz): 8.31 (s, 1H, H-2′), 7.89 (s, 1H, H-8′), 4.59 (dd, 1H, H-8, 7.8, 14.3), 4.52 (dd, 1H, H-8, 8.4, 14.3), 4.33 (brs, 4H, H-1″ +TFA), 4.14–4.08 (m, 2H, H-5, H-2), 3.19 (t, H-2″ in free *N*-Me-Piperazine, 4.5), 2.92 (dd, 1H, H-3, 8.1, 14.5), 2.69 (br.t, H-2″ in free *N*-Me-Piperazine, 4.5), 2.57 (t, 4H, H-2″, 4.5), 2.37–2.27 (m, 6H, H-1, H-4, H-7, CH_3_), 2.15 (brdt, 1H, H-3, 4.7, 14.5), 2.03 (ddd, 1H, H-6, 5.0, 10.0, 13.4), 0.92 (brd, 1H, H-6, 1.8, 13.4), ^13^C-NMR (CDCl_3_, δ ppm): 153.78 (C-6′), 152.22 (C-2′), 150.82 (C-4′), 138.99 (C-8′), 119.98 (C-**1**′), 69.27 (C-5), 60.56 (C-2), 54.98 (C-2″), 51.82 (CH2, C-2″ in free *N*-Me-Piperazine), 48.90 (C-7), 47.87 C-1), 46.00 (C-4), 45.74 (CH_3_-N), 44.78 (C-1″), 43.51 (CH2, C-1″ in free *N*-Me-Piperazine), 42.45 (C-8), 39.57 (C-6), 32.74 (C-3), the molecule crystalized with ~0.5 molecule of *N*-me-piperazine. After washing with water, the signals of free N-methyl piperazine disappeared.

#### 3.1.12. Synthesis of (1*S*,4*S*,5*S*,7*R*)-5-Chloro-7-((6-((4-methylpiperazin-1-yl)amino)-9*H*-purin-9-yl)methyl)bicyclo[2.2.1]heptan-2-ol, **6g**

Compound **5** (0.8 mmol), 4-methy-1-amino-piperazine (0.22 mL, 1.83 mmol), ethanol (7 mL), Et_3_N (0.6 mL) stirred for 72 h at rt, TLC (I, R_f_ = 0.08). LPC (III + 0.5 mL Et_3_N/100 mL system) gave 193 mg (49.2%) of pure compound **6g** as a foam, [α]_D_ = −14.3° (1% EtOH), IR: 3374brs, 3211s, 3073m, 2961s, 2859m, 1711w, 1607vs, 1592vs, 1516m, 1444m, 1410s, 1334m, 1299s, 1263s, 1237m, 1087m, 1008m, 992m, 796w, 649m, ^1^H-NMR (DMSO-*d*_6_, δ ppm, *J* Hz): 8.71 (s, 1H, NH), 8.22, 8.20 (s, 1H, H-2′, H-8″), 4.59 (dd, 1H, H-8, 10.0, 14.0), 4.32 (dd, 1H, H-8, 5.1, 14.0), 4.15 (dd, 1H, H-2, 3.7, 8.0), 3.84 (m, 1H, H-5), 2.84 (m, 4H, H-1′′), 2.74 (dd, 1H, H-3, 8.0, 14.0), 2.47–2.40 (m, 5H, H-4, 4H-2″), 2.35 (d, 1H, H-1, 4.6), 2.28–2.15 (m, 5H, H-3, H-7, CH_3_), 2.10 (m, 2H, H-4, H-6), 0.83 (dd, 1H, H-6, 2.2, 13.7), ^13^C-NMR (DMSO-*d*_6_, δ ppm): 172.66 (C-6′), 153.59 (C-4′), 152.40 (C-**2**′), 140.89 (C-**8**′), 117.66 (C-5′), 67.94 (C-5), 61.32 (C-2), 54.44 (C-1″), 48.94 (C-7), 48.36 C-1), 45.58 (CH_3_N), 45.55 (C-2″), 45.31 (C-4), 41.67 (C-8), 38.75 in DMSO (C-6), 32.18 (C-3).

#### 3.1.13. Synthesis of (1*S*,4*S*,5*S*,7*R*)-5-Chloro-7-((6-morpholino-9*H*-purin-9-yl)methyl)bicyclo[2.2.1]heptan-2-ol, **6h**

Compound **5** (0.8 mmol), morpholine (4 mL) stirred for 72 h at rt, TLC (I, R_f_ = 0.53). By crystallization of the crude product, 283.5 mg (66.1%) of pure compound **6h**, crystallized with two molecules of morpholine/mol of **6h**, were obtained as needless, mp 162.3–163.2 °C [LPC (I) of the mother liquors gave 75 mg (25.8%) of pure **6h**, total yield 91.9%], [α]_D_ = −15.8° (1% THF), IR: ^1^H-NMR (CDCl_3_, δ ppm, *J* Hz): 9.9 (brs, 1H, NH), 8.34 (s, 1H, H-2′), 7.90 (s, 1H, H-8′), 4.62 (dd, 1H, H-8, 7.6, 14.0), 4.54 (dd, 1H, H-8, 8.3, 14.0), 4.30 (brs, 4H, H-1″), 4.17–4.09 (m, 2H, H-5, H-2), 3.99 (t, 4H, H-1″ 4.6,), 3.82 (t, 4H, H-2″, in free morpholine, 4.4), 3.23 (t, 8H, H-1″, 4.6), 2.93 (dd, 1H, H-3, 8.1, 14.4), 2.38 (d, 1H, H-1, 5.0), 2.34 (d, 1H, H-4, 5.9), 2.30 (t, 1H, H-7, 8.3), 2.17 (brdt, 1H, H-3, 4.1, 14.4), 2.06 (ddd, 1H, H-6, 5.0, 10.1, 13.6), 0.92 (brdd, 1H, H-6, 2.4, 13.6), ^13^C-NMR (CDCl_3_, δ ppm): 153.90 (C-6′), 152.24 (C-2′), 150.82 (C-4′), 138.99 (C-8′), 119.98 (C-5′), 69.44 (C-5), 67.01 (C-2″), 63.73 (C-2″ in morpholine), 60.51 (C-2), 48.91 (C-7), 47.88 C-1), 45.71 (C-4), 45.56 (C-1″ in morpholine), 43.24 (C-1″), 42.46 (C-8), 39.55 (C-6), 32.72 (C-3). The molecule crystalized with ~2 molecule of morpholine. After washing NaOH in D_2_O, the proton signals of free morpholine moved to 3.65 and 2.87 ppm, both as triplet with *J* = 4.7 Hz.

#### 3.1.14. Synthesis of (1*S*,4*S*,5*S*,7*R*)-5-Chloro-7-((6-((4-hydroxyphenethyl)amino)-9*H*-purin-9-yl)methyl)bicyclo[2.2.1]heptan-2-ol, **6i**

Compound **5** (0.4 mmol), tyramine hydrochloride (0.8 mmol, 140 mg), ethanol (3.5 mL), Et_3_N (0.3 mL) were stirred for six days at rt, TLC (I, R_f_ = 0.49). LPC (I) gave 151.5 mg (91.5%) of pure compound **6i** as a foam, [α]_D_ = 6.4° (1% EtOH), IR: 3268br, 2934m, 2805w, 2678w, 2593w, 1890w, 1616vs, 1542w, 1512m, 1482m, 1446m, 1338m, 1298m, 1228s, 1170w, 1082m, 1050w, 900w, 648w, ^1^H-NMR (DMSO-*d*_6_ + trases CDCl_3_, δ ppm, *J* Hz): 9.18 (s, 1H, OH), 8.23 (s, 1H, H-2′), 8.17 (s, 1H, H-8′), 7.72 (brs, 1H, NH), 7.05 (d, 2H, H-*o*, 7.8), 6.98 (d, 2H, H-*m*, 7.8), 4.71 (brs, 1H, 5-OH), 4.59 (dd, 1H, H-8, 10.0, 14.0), 4.32 (dd, 1H, H-8, 4.7, 14.0), 4.14 (m, 1H, H-2), 3.84 (m, 1H, H-5), 3.63 (brs, 2H, CH_2_-NH), 2.79 (t, 2H, CH_2_-Ph, 7.6), 2.73 (dd, 1H, H-3, 7.8, 13.9), 2.33 (d, 1H, H-1, 4.4), 2.26–2.21 (m, 2H, H-3, H-7), 2.03 (brs, 1H, H-4), 1.98 (m, 1H, H-6), 0.81 (d, 1H, H-6, 13.2), ^13^C-NMR (DMSO-*d*_6_, δ ppm): 155.60 (C-*p* Ar), 154.42 (C-6′), 152.48 (C-2′), 148.70 (C-4′), 140.47 (C-8′), 129.52 (2C-C-*o*), 118.94 (C-5′), 115.11 (2C, C-*m*), 67.95 (C-5), 61.30 (C-2), 48.95 (C-7), 48.35 C-1), 45.30 (C-4), 41.70 (C-8), 39.47 (C-6), 32.19 (C-3).

#### 3.1.15. Synthesis of (1*S*,4*S*,5*S*,7*R*)-5-Chloro-7-((6-((4-hydroxyphenethyl)amino)-9H-purin-9-yl)methyl)bicyclo[2.2.1]heptan-2-ol, **6j**

Compound **5** (0.4 mol), 4-methoxyphenethylamine (121 mg, 0.8 mmol), ethanol (3,5 mL), Et_3_N (0.3 mL) were stirred for six days at rt, TLC (I, R_f_ = 0.68). LPC (I) gave 134.4 mg (78.5%) of pure compound **6j** as a foam, [α]_D_ = −4.0° (1% EtOH), IR: 3400m, 3293m, 2997m, 2939m, 2872m, 2834m, 1969w, 1704w, 1614vs, 1576s, 1510m, 1463m, 1440m, 1333s, 1294s, 1236s, 1079m, 1003w, 896w, 644w, NMR (CDCl_3_, δ ppm, *J* Hz): 8.39 (s, 1H, H-2′), 7.86 (s, 1H, H-8′), 7.72 (brs, 1H, NH), 7.17 (d, 2H, H-*o*, 8.2), 6.85 (d, 2H, H-*m*, 8.2), 5.90 (brs, 1H, 5-OH), 4.62 (dd, 1H, H-8, 8.0, 14.3), 4.55 (dd, 1H, H-8, 8.3, 14.0), 4.17–4.10 (m, 2H, H-2, H-5), 3.89 (brs, 2H, CH_2_-NH), 3.79 (s, 3H, CH_3_O), 2.96 (m, 1H, H-3), 2.93 (t, 2H, CH_2_-Ph, 7.0), 2.40 (d, 1H, H-1, 4.4), 2.18 (brdt, 1H, H-3, 14.6), 2.36–2.30 (m, 2H, H-4, H-7), 2.07 (ddd, 1H, H-6, 5.1, 9.9, 13.7), 0.91 (brd, 1H, H-6, 13.7), ^13^C-NMR (CDCl_3_, δ ppm): 158.24 (C-*p* Ar), 154.79 (C-6′), 153.06 (C-2′), 149.16 (C-4′), 140.01 (C-8′), 130.80 (C-1Ar), 130.80 (2C-C-*o*), 129.75 (C-*o*), 119.70 (C-5′), 114.03 (2C, C-*m*), 69.40 (C-5), 60.54 (C-2), 55.26 CH_3_O), 49.05 (C-7), 47.95 C-1), 45.73 (C-4), 42.55 (C-8), 42.10 (CH_2_NH), 39.51 (C-6), 35.00 (CH_2_-Ph), 32.71 (C-3).

#### 3.1.16. Synthesis of (1*S*,4*S*,5*S*,7*R*)-5-Chloro-7-((6-(phenethylamino)-9*H*-purin-9-yl)methyl)bicyclo[2.2.1]heptan-2-ol, **6k**

Compound **5** (0.4 mmol), 2-phenylethylamine (1 mL), were stirred for six days at rt, TLC (I, R_f_ = 0.54). LPC (I) gave 124 mg (77.9%) of pure compound **6k** as a foam, [α]_D_ = −25.8° (1% EtOH), IR: 3268brm, 3029m, 2952m, 1616vs, 1538w, 1483m, 1448m, 1332m, 1296m**,** 1226m**,** 1082m, 1001m, 900w, 647w, ^1^H-NMR (CDCl_3_, δ ppm, *J* Hz): 8.39 (s, 1H, H-8′), 7.85 (s, 1H, H-2′), 7.33–7.22 (m, 5H, H-Ar), 6.01 (s, 1H, NH), 4.62 (dd, 1H, H-8, 8.1, 14.2), 4.54 (dd, 1H, H-8, 8.1, 14.2), 4.17–4.08 (m, 2H, H-5, H-2), 3.97–3.86 (m, 2H, H-1′), 2.99 (m, 1H, H-3), 2.95 (t, 2H, H-2″, 8.3, 16.0), 2.40 (d, 1H, H-1, 4.5), 2.35–2.30 (m, 2H, H-4, H-7), 2.30 (t, 1H, H-7, 8.3), 2.18 (brdt, 1H, H-3, 4.1, 17.6), 2.06 (ddd, 1H, H-6, 5.0, 10.1, 13.8), 0.93 (brd, 1H, H-6, 2.2, 13.8), ^13^C-NMR (CDCl_3_, δ ppm): 154.75 (C-6′), 153.05 (C-2′), 148.91 (C-4′), 139.92 (Cq-Ar), 138.82 (C-8′), 128.80 (2C-C-*m*), 128.60 (2C, C-*o*), 119.66 (C-5′), 69.31 (C-5), 60.54 (C-2), 49.03 (C-7), 47.95 C-1), 45.72 (C-4), 42.56 (C-8), 41.80 (C-1″), 39.47 (C-6), 35.88 (C-2″), 32.72 (C-3).

##### General Procedure for Obtaining the 6-Alkoxy-purine Analogs **7a** and **7b**

Sodium (25 mg) was reacted with methanol or ethanol (3.5 mL), compound **5** (0.4 mmol) was added and stirred for four days monitoring the end of the reaction by TLC (I). The base was neutralized (1 M HCl), volatiles were removed under reduced pressure and the residue was purified by LPC (solvent system I).

#### 3.1.17. Synthesis of (1*S*,4*S*,5*S*,7*R*)-5-Chloro-7-((6-methoxy-9*H*-purin-9-yl)methyl)bicyclo[2.2.1]heptan-2-ol, **7a**

Compound **5** (0.4 mmol), methanol (3.5 mL), Na (25 mg) were stirred for four days at rt, TLC (I, R_f_ = 0.50). By LPC, 87.6 mg (70.9%) of crystallized **7a**, mp 196.2–197.2 °C (CH_2_Cl_2_-Hexane), [α]_D_ = 25.2° (1% EtOH), IR: 3328m, 3072w, 3006w, 2964w, 2940w, 2874w, 2142w, 1812w, 1596vs, 1478m, 1417w, 1374m, 1315s, 1231s, 1083s, 1003m, 960m, 892w, 647w, ^1^H-NMR (DMSO-*d*_6_, δ ppm, *J* Hz): 8.54 (s, 1H, H-2′), 8.44 (s, 1H, H-8′), 4.72 (d, 1H, OH, 2.8), 4.70 (dd, 1H, H-8, 10.3, 14.2), 4.42 (dd, 1H, H-8, 5.2, 14.2), 4.15 (brdd, 1H, H-2, 3.1, 8.0), 4.09 (s, 3H, CH_3_), 3.85 (m, 1H, H-5), 2.74 (dd, 1H, H-3, 8.0, 14.2), 2.34 (d, 1H, H-1, 4.6), 2.25 (d, 1H, H-3, 14.2), 2.23 (m, 1H, H-7), 2.04–1.94 (m, 2H, H-6, H-4), 0.82 (brdd, 1H, H-6, 2.2, 13.3), ^13^C-NMR (DMSO-*d*_6_, δ ppm): 160.27 (C-6′), 152.06 (C-2′), 151.50 (C-4′), 136.00 (C-8′), 120.45 (C-5′), 67.97 (C-5), 62.25 (C-2), 53.90 (OCH_3_), 48.74 (C-7), 48.41 C-1), 45.30 (C-4), 42.16 (C-8), 38.90 (C-6) in DMSO, 32.17 (C-3).

#### 3.1.18. Synthesis of (1*S*,4*S*,5*S*,7*R*)-5-Chloro-7-((6-ethoxy-9*H*-purin-9-yl)methyl)bicyclo[2.2.1]heptan-2-ol, **7b**

Compound **5** (0.4 mmol), ethanol (3.5 mL), Na (25 mg) were stirred for four days at rt, TLC (I, R_f_ = 0.59). By LPC, 108 mg (83.6%) **7b** were obtained as a foam, [α]_D_ = 11.6° (1% EtOH), IR: 3317brm, 2962m, 2728w, 1907w, 1598vs, 1508w, 1456s, 1412m, 1317s, 1223s, 1145w, 1053s, 1002m, 900w, 648m, ^1^H-NMR (CDCl_3_, δ ppm, *J* Hz): 8.51 (s, 1H, H-2′), 8.05 (s, 1H, H-8′), 4.73–4.59 (m, 3H, H-8, 2H-1″), 4.58 (dd, 1H, H-8, 7.9, 14.8), 4.18–4.11 (m, 1H, H-2, H-5), 3.85 (m, 1H, H-5), 2.96 (dd, 1H, H-3, 8.0, 14.6), 2.41 (d, 1H, H-1, 4.7), 2.25 (d, 1H, H-3, 14.2), 2.35–2.30 (m, 2H, H-4, H-7), 2.21 (brt, 1H, H-3, 4.2, 14.9), 2.07 (ddd, 1H, H-6, 5.0, 10.0, 13.8), 0.95 (brdd, 1H, H-6, 2.2, 13.8), ^13^C-NMR (CDCl_3_, δ ppm): 160.88 (C-**6**′), 152.04 (C-**4**′), 151.96 (C-**2**′), 142.32 (C-**8**′), 121.44 (C-**5**′), 69.43 (C-5), 63.16 (C-1″), 60.44 (C-2), 49.08 (C-7), 48.03 C-1), 45.76 (C-4), 42.82 (C-8), 36.51 (C-6), 32.67 (C-3), 14.52 (C-2″).

### 3.2. Anti-Viral Testing of the Compounds Viruses and Cells

Influenza virus (Flu A, strain A/Puerto Rico/8/34 (H1N1)), human adenovirus type 5 (AdV5) and herpes simplex virus type 1 (HSV-1) were obtained from the collection of viruses of the Pasteur Institute (St. Petersburg, Russia). Prior to the experiment, the influenza virus was propagated in the allantoic cavity of 10–12 days old chicken embryos for 48 h at 36 °C. AdV5 and HSV-1 were grown in Vero cells for three days at 36 °C and 5% CO_2_. Infectious titers of FluA was determined in MDCK cells (ATCC # CCL-34) cells, of AdV5 and HSV-1—in Vero cells (ATCC #CCL-81). Cells were seeded into 96-wells plates in Eagle′s minimal essential medium (MEM).

#### 3.2.1. Cytotoxicity Assay.

The microtetrazolium test (MTT) was used to study the cytotoxicity of the compounds (Mossman, 1983). Briefly, a series of three-fold dilutions of each compound (1000–4 µg/mL) in MEM were prepared. Cells were incubated for 48 h (MDCK) or 72 h (Vero) at 36 °C in 5% CO_2_ in the presence of the dissolved substances. The cells were then washed twice with saline, and a solution of 3-(4,5-dimethylthiazolyl-2) 2,5-diphenyltetrazolium bromide (ICN Biochemicals Inc., Aurora, OH, USA) (0.5 µg/mL) in saline was added to the wells. After 1 h of incubation, the wells were washed, and the formazan residue was dissolved in DMSO (0.1 mL per well). The optical density of cells was then measured on a Victor^2^1440 multifunctional reader (Perkin Elmer, Turku, Finland) at a wavelength of 535 nm and plotted against the concentration of the compounds. Each compound concentration was tested in three parallels. The 50% cytotoxic dose (CC_50_) of each compound (i.e., the compound concentration that causes the death of 50% of cells in a culture, or decreases the optical density twice as compared to the control wells) was calculated from the data obtained.

#### 3.2.2. Virus Titration.

The compounds in appropriate concentrations were dissolved in MEM with 1 µg/mL trypsin for Flu A and without additives for AdV5 and HSV-1, and incubated with cells for 1 h at 36 °C. The cell culture was then infected with either virus (moi 0.01). The plates were incubated for 48 h (Flu A) or 72 h (AdV5 and HSV-1) at 36 °C in the presence of 5% CO_2_. The activity of the viruses was evaluated by the MTT test, as described above, as its ability to decrease cell viability. Each concentration of the compounds was tested in triplicate. The antiviral activity of the compounds was estimated by the protection of cells from virus-induced death as compared with the control. The 50% inhibiting concentration (IC_50_) of the drug, that is, the concentration at which 50% of cells were protected, compared to control wells, and the selectivity index (the ratio of CC_50_ to IC_50_) were calculated from the data obtained.

Antiviral assays with EV71, YFV, and Chikungunya where performed as described earlier [46,47,48].

## 4. Conclusions

New 1′homocarbanucleoside analogs with pyrimidines, 6-chloropurine, adenine and 6-substituted adenine, 6-methoxy and ethoxy-purine as nucleobase were synthesized from an optically active 5-*endo*-hydroxy-8-hydroxymetil- bicyclo[2.2.1]heptane skeleton as sugar moiety. The 1′-homonucleosides with uracil, 5-fluorouracil, thymine, cytosine, and 6-chloropurine as nucleobases were synthesized by a selective Mitsunobu reaction on the primary hydroxymethyl group in the presence of 5-*endo*-hydroxyl group of the norbornane starting compound. Adenine and 6-substituted adenine homonucleosides were obtained by substitution of the 6-chlorine atom of the key intermediate **5** with ammonia and selected amines, and 6-methoxy- and 6-ethoxy substituted purine analogs by reaction with the corresponding alkoxides. The compounds were inactive against entero-, yellow fever, Chikungunya and adenoviruses. Two compounds (**6j** and **6d**) have lower IC_50_ (15 ± 2 and 21 ± 4 µM) and one compound has IC_50_ similar to that of acyclovir (28 ± 4 µM).

These suggest that the bicyclo[2.2.1]heptane fragment can be effectively used instead of ribose or deoxyribose moiety to give nucleoside analogs with virus-inhibiting activity. The exact mechanism(s) of their anti-viral activity should be further studied in order to optimize the structure to create derivatives with low toxicity and high selectivity against specific viruses and with broad range of activity.

## 5. Patents

A patent pending dealing with the synthesis of the compounds was also filled.

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
