# Peer review of "New HSV-1 Anti-Viral 1′-Homocarbocyclic Nucleoside Analogs with an Optically Active Substituted Bicyclo[2.2.1]Heptane Fragment as a Glycoside Moiety"

_molecules, 2019, doi:10.3390/molecules24132446_

Round 1

Reviewer 1 Report

This manuscript describes the synthesis and biological activity of a variety of 1’homocarbanucleoside analogues with an optically active substituted bicyclo[2.2.1]heptane skeleton as sugar moiety.  Author found the three active compounds to herpes virus with similar IC50 value to acyclovir.  These results are very interesting, but this manuscript is a lack of the discussion. Authors performed the docking studies between their synthesized ligand and thymidine kinase as a target of the HSV-1 inhibitors.  Authors should discuss the relationship between the score in docking studies and IC50, not only 6j, but also other compounds. Before publishing, I would comment the following points:

1)      In the docking study, they showed score and should explain the meaning of score.

2)      Mitsunobu reaction to alkylate the pyrimidine bases were lower yields than that of purine bases.  Authors should discuss on it.  For example, in the reaction to the pyrimidine, the bi-product which reacted to another hydroxyl group was produced or not.

3)      Authors should depict the reagents and condition each step in scheme 1 and 2.

4)      The compounds, 6f, 6d and 6j showed similar activity to herpes virus with acyclovir, but increasing cytotoxicity.  Authors should comment to it.

5)      This manuscript contains some unclear statements.  I suggest that the authors seek the advice of someone with a good knowledge of English.

Minor points

1)      Authors should insert the unit of concentration in Table 2.

2)      In experimental section, authors depict “13C-RMN” and should correct “13C-NMR”.

3)      Authors did not describe the Mass spectrum for all synthesized compounds and should describe them.

Author Response

Answers to reviewer # 1 (Thank you for your valuable observations):

General observation: “Authors should discuss the relationship between the score in docking studies and IC50, not only 6j, but also other compounds”. A phrase was introduced on page 5, lines 156-157. The comments: 1.“In the docking study, they showed score and should explain the meaning of score”. Usually the meaning of the docking score is known, but we introduced on page 5, in the footer of the Table, the following phrase: The docking score (PLANTPLP score) is a function described in Korb et al. [42]. For a strong binding, the score has a negative value; fore weak or non-existing binding the score has a less negative or even positive value.   2. “Mitsunobu reaction to alkylate the pyrimidine bases were lower yields than that of purine bases.  Authors should discuss on it.  For example, in the reaction to the pyrimidine, the bi-product which reacted to another hydroxyl group was produced or not”. It was not in our intention to isolate the secondary compounds. From our experience, in the Mitsunobu reaction of the protected diol 2 with 5-FU (5-fluoror uracil) and T (timine), we isolated O2,O4- and N1,O4-bis-alkylated secondary nucleosides, the results being published in the paper Tanase, C. et al. [43]. On page 7 we introduced the following comment: The reduced yield for the pyrimidine analogues is attributed to the formation of O2,O4 - N1,O4 - bis-akylated secondary compounds, as we observed previously [43] for alkylation of the diol 2 protected at the primary hydroxyl as benzoate; in this case could be also alkylated compounds to both hydroxyls oh the diol 2. All secondary compounds had higher mobility on TLC, in the domain of triphenylphosphine oxide as Rf, and were not isolated pure for characterization.   3. “Authors should depict the reagents and condition each step in scheme 1 and 2” In Schemes 1 and 2 we introduced the numbers of the steps and in the capture the reaction conditions. Scheme 1. Synthesis of pyrimidine N’-homocarbanucleoside analogues 4a-4c: 1) 1, NaBH4, MeOH, -16 ˚C [36]; 2) 5FU, U or T, Ph3P, THF 0 ˚C, DIAD added dropwise, 1h stirring, 2 in THF added in 50 min, stirring overnight; for 4d, N4-benzoyl cytosine and crude product hydrolyzed with MeONa/MeOH, rt, overnight. Scheme 2. Synthesis of 6-purine substituted 1’-homocarbanucleoside analogues: 5, 6a-6k, 7a-7b: 1) 6-Cl-Pu, Ph3P in THF, DIAD, -6˚ C, 1h, 2 in THF added in 1h, then stirring overnight; 2) for 6a: 5, 25 % NH4OH, MeOH, 80-90 ˚C (pressure vessel) 24h; for 6b-6d, 6h, 6k: 5, amine, rt, 72h - 6 days; for compounds 6e-6g, 6i-6g: 5, amine, EtOH, EtN3, 72h (6f, 6g) to 6 days (6i, 6g), rt, (60 ˚C for 6e). 4. “The compounds, 6f, 6d and 6j showed similar activity to herpes virus with acyclovir, but increasing cytotoxicity.  Authors should comment to it.” We absolutely agree with you. We indicated this issue specifically in the text and briefly discussed it (p.9). 5. “This manuscript contains some unclear statements.  I suggest that the authors seek the advice of someone with a good knowledge of English.” We corrected the manuscript. Minor points: 1. “Authors should insert the unit of concentration in Table 2.” Thank you. We added the units (µM). 2. In experimental section, authors depict “13C-RMN” and should correct “13C-NMR”. We corrected (Thank you very much for the observation). 3. “Authors did not describe the Mass spectrum for all synthesized compounds and should describe them.” At this step we had a problem with the mass spectrometer and there are no funds presently for the cost of the reparation. The bicyclo[2.2.1]heptane fragment was fully characterized previously, including the X-ray crystallography for a few compounds.

Reviewer 2 Report

In this manuscript, Alexander et. al. synthesized a series of novel 1’-homocarbocyclic purine and pyrimdine nucleoside analogues from a bicyclo[2.2.1]heptane scaffold via the selective Mitsunobu reaction, and the candidates were chosen on the basis of preceding molecular docking simulation. The antiviral assay revealed that all reported compounds are inactive against entero-, yellow fever, Chikungunya and adeno- viruses, but some showed promising activity against herpes virus with SI higher than 10.

Comments:

1. The introduction part is far too lengthy and fragmentated, please try to provide a concise overview of the known knowledge related to the presented research work.

2. The antiviral assay results on EV71, YFV and Chikungunya must be provided either in the main body of the manuscript or in the supporting information.

3. The “References” part needs to be arranged according to the author guideline.

4. There is still some space for the authors to improve the professional language skill.

Author Response

Answer to  the reviewer # 2:

The comments:

1. “The introduction part is far too lengthy and fragmentated, please try to provide a concise overview of the known knowledge related to the presented research work”.

We improved the introduction in the manuscript.

2. “The antiviral assay results on EV71, YFV and Chikungunya must be provided either in the main body of the manuscript or in the supporting information.”

We introduced Table 3 in the manuscript and switched the order of part 1 with part 2 (lines 215-219). The phrase from lines 220-221 was moved to lines 263-264.

3. The “References” part needs to be arranged according to the author guideline.

The spaces were corrected. (Thank you very much for observation).

4. There is still some space for the authors to improve the professional language skill.

The manuscript was corrected.

@font-face { font-family: "Times New Roman"; }@font-face { font-family: "宋体"; }@font-face { font-family: "Calibri"; }@font-face { font-family: "Palatino Linotype"; }@font-face { font-family: "SimSun"; }p.MsoNormal { line-height: 107%; font-family: Calibri; color: rgb(0, 0, 0); }p.NewStyle15 { font-family: "Palatino Linotype"; color: rgb(0, 0, 0); font-weight: bold; }p.p { line-height: 107%; font-family: "Times New Roman"; }span.msoIns { text-decoration: underline; color: blue; }span.msoDel { text-decoration: line-through; color: red; }div.Section0 { }

Reviewer 3 Report

Dear Authors,

The manuscript focuses on the antiviral activity of your compounds. Therefore, I would advise changing the title of your manuscript to include information about their antiviral activity.

I cannot agree with the statement in line 223 – “It is known that the target for the best known anti-herpes compound acyclovir is virus-encoded thymidine kinase”. Acyclovir (ACV) is phosphorylated by viral thymidine kinase (TK), and its triphosphate form is incorporated into the DNA resulting in chain termination. Therefore, TK is not the target of ACV responsible for its antiviral activity. Other nucleic acid synthesis inhibitors may have different mode of action. For example, sorivudine triphosphate blocks viral DNA replication by inhibiting DNA polymerase activity, but it is not incorporated into elongating viral DNA.

In the next line (224) – “Probably, the compounds we studied interfere with this enzyme, and for this reason are ineffective against influenza virus who does not encode its own kinase”. As was mentioned before, TK is not the target for nucleic acid synthesis inhibitors. The word “interfere” may suggest that your compounds inhibit TK, which is not supported by the data you present.  

Table 2. – Please include the CC50 and IC50 units in the description (footer) of the table.

Search the manuscript for IC50 and write “50” in subscript (IC50).

Line 540 – what do you mean by “either virus viruses”?

Line 541 – the MMT test you have used doesn’t allow for the measurement of “infection activity of the viruses”. The only thing you can assess using MTT in your assay is the relationship between cytopathic effect (CPE) formation, measured as the decrease of cell viability, and the concentration of particular compound.

Some of your compounds, especially 6c, 6d, 6f, 6j showed promising antiviral activity. It would greatly increase the scientific value of the manuscript to assess the antiviral activity of those compounds on different steps of replication cycle of the HSV-1. At least you could use end point dilution assay to measure the virus titer after incubation of HSV-1 infected cell line with your compounds.

Author Response

Answer to the reviewer # 3 (thank you for your valuable observations):

The comments: -The manuscript focuses on the antiviral activity of your compounds. Therefore, I would advise changing the title of your manuscript to include information about their antiviral activity. We introduced in the title HSV-1 Anti-viral, and the title is: “New HSV-1 Anti-viral 1’-Homocarbocyclic Nucleoside Analogues with an Optically Active Substituted Bicyclo[2.2.1]Heptane Fragment as a Glycoside Moiety” -I cannot agree with the statement in line 223: “It is known that the target for the best known anti-herpes compound acyclovir is virus-encoded thymidine kinase”. Acyclovir (ACV) is phosphorylated by viral thymidine kinase (TK), and its triphosphate form is incorporated into the DNA resulting in chain termination. Therefore, TK is not the target of ACV responsible for its antiviral activity. Other nucleic acid synthesis inhibitors may have different mode of action. For example, sorivudine triphosphate blocks viral DNA replication by inhibiting DNA polymerase activity, but it is not incorporated into elongating viral DNA. -In the next line (224) “Probably, the compounds we studied interfere with this enzyme, and for this reason are ineffective against influenza virus who does not encode its own kinase”. As was mentioned before, TK is not the target for nucleic acid synthesis inhibitors. The word “interfere” may suggest that your compounds inhibit TK, which is not supported by the data you present. You are right. We corrected the text accordingly (lines 236-244). Table 2. – Please include the CC50 and IC50 units in the description (footer) of the table. -Search the manuscript for IC50 and write “50” in subscript (IC50). We corrected all the cases. -Line 540 – what do you mean by “either virus viruses”? This was a technical mistake. Thank you for indicating this, we removed the inappropriate word. -Line 541 – the MMT test you have used doesn’t allow for the measurement of “infection activity of the viruses”. The only thing you can assess using MTT in your assay is the relationship between cytopathic effect (CPE) formation, measured as the decrease of cell viability, and the concentration of particular compound. We corrected the sentence. Right, this was not the evaluation of the ineffectiveness of the virus. -Some of your compounds, especially 6c, 6d, 6f, 6j showed promising antiviral activity. It would greatly increase the scientific value of the manuscript to assess the antiviral activity of those compounds on different steps of replication cycle of the HSV-1. At least you could use end point dilution assay to measure the virus titer after incubation of HSV-1 infected cell line with your compounds. In the framework of this study our goal was to evaluate the anti-viral potential of novel nucleoside analogs with bicyclo[2.2.1]heptane skeleton as sugar moiety against viruses of phylogenically distinct groups; and we did not plan for the detailed study of their mode of action. However, the issues of target identification and mechanism of activity are important for complex drug development, although another specific study should be devoted to this. We added the sentence describing our further plans regarding the study of the mechanism of action and target identification (p.8).

Round 2

Reviewer 2 Report

This revised version stands as a comprehensive study on a series of novel 1’-homocarbocyclic purine and pyrimdine nucleoside analogues in pursuit for lead antiviral candidates. I am pleased to see that significant improvement has been made, and therefore I recommend it for publication in Molecules if one thing could be addressed more carefully.

Comments:

1. There are still too much details in the introduction part; at least I would like to suggest: 1) Figure 1 and Figure 2 can be combined; 2) those detailed examples from line 77 to 96 should be removed or heavily concentrated, for an instance, only keeping each compound number (V, VI…) with the corresponding reference(s).

Author Response

Dear reviewer #2,

Thank you for your very useful observations.

About your last two observations:

1) You have recommended to combine the Fig. 1 with Fig. 2.

 Sure, this is easy to be done, but in the actual form the formulas in the figures are in the nearest position to the place where are mentioned in the text, and this is useful for the reader; and when the Fig. appear on a different page then the text describing it, is difficult for readers.

and

2)those detailed examples from line 77 to 96 should be removed or heavily concentrated, for an instance, only keeping each compound number (V, VI…) with the corresponding reference(s).

My intention was to give some minimal data about the (antiviral or anticancer) activity of the most active compounds to be useful in showing the state of the art in the field; in this context, our results, not better or worse, could be integrated in these published data and evaluated by readers in reading and use for their personal research. In my previous papers (for ex. Bioorg. Med. Chem., 2015, 23, 6346, New J. Chem, 2019, 43, 7582), in the introduction, I introduced also minimal data about the compounds/procedures, in the same idea, and the papers were accepted. 

I corrected the manuscript as you recommended at point 2, though I remain to my opinion that the minimal data (in the length of  <5 rows) could remain.

I remain yours sincerely,

Constantin Tanase